# GRAPH LEARNING NETWORK: A STRUCTURE LEARNING ALGORITHM

## ABSTRACT

Graph prediction methods that work closely with the structure of the data, e.g., graph generation, commonly ignore the content of its nodes. On the other hand, the solutions that consider the node's information, e.g., classification, ignore the structure of the whole. And some methods exist in between, e.g., link prediction, but predict the structure piece-wise instead of considering the graph as a whole. We hypothesize that by jointly predicting the structure of the graph and its nodes' features, we can improve both tasks. We propose the Graph Learning Network (GLN), a simple yet effective process to learn node embeddings and structure prediction functions. Our model uses graph convolutions to propose expected node features, and predict the best structure based on them. We repeat these steps sequentially to enhance the prediction and the embeddings. In contrast to existing generation methods that rely only on the structure of the data, we use the feature on the nodes to predict better relations, similar to what link prediction methods do. However, we propose an holistic approach to process the whole graph for our predictions. Our experiments show that our method predicts consistent structures across a set of problems, while creating meaningful node embeddings.

## 1 INTRODUCTION

Data is organically structured (and can be represented as a graph) as relations exist between its elements (nodes on such graph), e.g., networks, images, proteins, etc. Hence, learning to predict this structure from its components plays an important role in understanding the data and the process that generated it. We consider the problem of predicting the structure of a given set of points (which we assume are the nodes of a graph) and an initial structure (connections of the points). Simultaneously, we aim to learn to predict these structures according to some prior information.

Predicting the structure and nodes' information of a graph is not a new task. Existing approaches tend to focus on one of these tasks, and ignore the properties of the other in their solutions. For instance, generative graph models (Grover et al., 2018; Li et al., 2018; Simonovsky & Komodakis, 2018; You et al., 2018) create graphs (mainly the structure) and ignore the features involved on this process. Node classification methods (Defferrard et al., 2016; Kipf & Welling, 2017; Lee et al., 2018; Zhang et al., 2018) work closely with the features of the nodes, but assume that the graph structure is fixed and given, both of which restrict the problems that can be solved. And link prediction methods (Grover & Leskovec, 2016; Kipf & Welling, 2016; Perozzi et al., 2014) are a compromise in between. However, they work looking at pairs of nodes at a time and, commonly, ignore the whole structure of the graph to make its predictions.

On the contrary, in this paper, we present a simple yet effective method to predict the structure of a given set of points, that we assume have an underlying graph structure, and create node embeddings of their original features that are more robust for further processing (e.g., classification). Our proposal comprises a two step repetitive process that obtains expected node embeddings, and then use them to obtain the best prediction of the structure given the information at that step. These steps are repeated on a refinement process, and encoded as layers in a neural network.

Our contribution is the definition of two prediction functions (for nodes' features and adjacency), that let us extract the most probable structure given a set of points and their feature embeddings, respectively. We also present a layer-wise architecture that define our iterative process and our prediction functions, and a learning framework that let us learn, on and end-to-end fashion, how

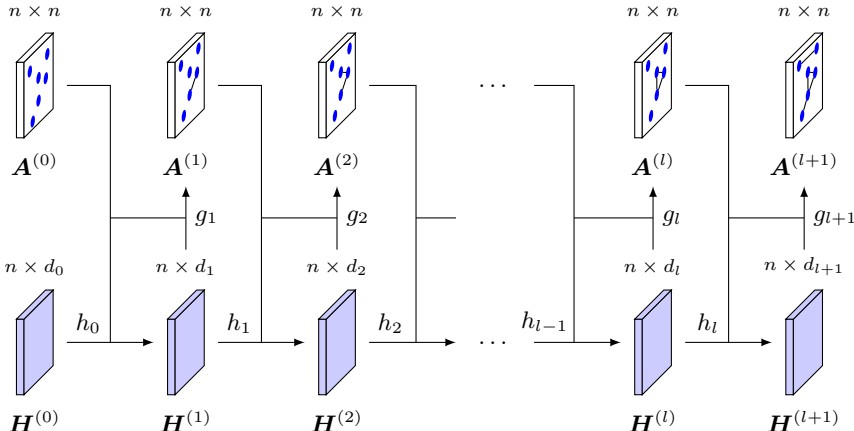

**Figure 1.** Our proposed method comprises two steps. First, a node embedding function, $h_l$, that predicts the expected features, $\boldsymbol{H}^{(l+1)}$, given our belief of the current structure, $\boldsymbol{A}^{(l)}$, and the existing features, $\boldsymbol{H}^{(l)}$. Then, a prediction function, $g_l$, of the most likely structure, $\boldsymbol{A}^{(l+1)}$, is applied based on the current features and our previous approximation of it. We alternate these steps for a given number of steps, and the resulting features and structure represent the predicted graph. Our objective is to learn the $h_l$ and $g_l$ functions based on a set of graphs such that we can predict similar structures.

to predict the structure given a family of graphs. We demonstrate that our proposed method can effectively extract relevant features and generate graph representations on diverse settings. Additionally, we introduce a synthetic dataset that contains patterns that can be controlled and mapped into graphs to evaluate the robustness of existing methods. We present more details regarding the dataset on Appendix A.

## 2 Graph Learning Network

Given a set of vertices $\mathbb{V} = \{\boldsymbol{v}_i\}$, such that every element $\boldsymbol{v}_i$ is a feature vector, we intend to predict its structure as a set of edges between the vertices, $\mathbb{E} = \{(\boldsymbol{v}_i, \boldsymbol{v}_j) : \boldsymbol{v}_i, \boldsymbol{v}_j \in \mathbb{V}\}$. In other words, we want to learn the edges of the graph $\mathcal{G} = (\mathbb{V}, \mathbb{E})$ that maximize the relations between the vertices given some prior patterns, i.e., a family of graphs.

To achieve this, we perform two alternating tasks for a given number of times (this defines our architecture), akin to an expectation-maximization process. At each step, we transform the nodes' features through convolutions on the graph (Kipf & Welling, 2017) to learn better representations to predict their structure. Then, we use these transformed features to predict the next structure, which is represented through an adjacency matrix. The learned convolutions on the graph represent a set of responses on the nodes that will reveal their relations. These responses are combined to create or delete connections between the nodes, and encoded into the adjacency matrix. The sequential application of these steps recover effective relations on nodes, even when trained on families of the graphs. We represent this process, which is illustrated in Fig. 1, through a neural network that is trained in an end-to-end fashion to learn, both, the convolution kernels and the structure-predicting functions.

### 2.1 Node Embeddings

At a given step, $l$, on the alternating process, we have the $d_l$ hidden features, $\boldsymbol{H}^{(l)} \in \mathbb{R}^{n \times d_l}$, of the $n$ nodes, and the set of edges (structure) encoded into an adjacency matrix $\boldsymbol{A}^{(l)} \in [0, 1]^{n \times n}$ that represent our graph. As introduced, our first step is to produce the features of the next step, $\boldsymbol{H}^{(l+1)}$, through the embedding function, $h_l$,

$$\boldsymbol{H}^{(l+1)} = h_l\left(\boldsymbol{H}^{(l)}, \boldsymbol{A}^{(l)}\right). \tag{1}$$

In our proposal, we intend to transform each node's features locally by using the information of its neighborhood on the graph. Hence, we use the convolutional graph operation proposed by Kipf & Welling (2017)

$$h_l\left(\boldsymbol{H}^{(l)}, \boldsymbol{A}^{(l)}\right) = \sigma_l\left(\tau\left(\boldsymbol{A}^{(l)}\right)\boldsymbol{H}^{(l)}\boldsymbol{W}^{(l)}\right), \tag{2}$$

where $\boldsymbol{W}^{(l)} \in \mathbb{R}^{d_l \times d_{l+1}}$ is the learnable weights of the convolution kernel for the $l$th step, $\sigma_l$ is an activation function, and $\tau(\cdot)$ is a symmetric normalization transformation of the adjacency matrix, defined by

$$\tau\left(\boldsymbol{A}^{(l)}\right) = \left(\hat{\boldsymbol{D}}^{(l)}\right)^{-\frac{1}{2}}\left(\boldsymbol{A}^{(l)} + \boldsymbol{I}_n\right)\left(\hat{\boldsymbol{D}}^{(l)}\right)^{-\frac{1}{2}}, \tag{3}$$

where $\hat{\boldsymbol{D}}^{(l)}$ is the degree matrix of the graph plus the identity, that is,

$$\hat{\boldsymbol{D}}^{(l)} = \boldsymbol{D}^{(l)} + \boldsymbol{I}_n, \tag{4}$$

where $\boldsymbol{D}^{(l)}$ is the degree matrix of $\boldsymbol{A}^{(l)}$, and $\boldsymbol{I}_n$ is the identity matrix of size $n \times n$. Unlike previous work (Kipf & Welling, 2017), we are computing convolutions that will have different neighborhoods at each step defined by the changing $\boldsymbol{A}^{(l)}$. In summary, this step allow us to learn a response function, defined by the weights $\boldsymbol{W}^{(l)}$ of the kernel, that embed the node's features into a suitable form to predict the structure of the graph.

## 2.2 ADJACENCY MATRIX PREDICTION

After obtaining the nodes embedding, $\boldsymbol{H}^{(l)}$, we predict the adjacency matrix, $\boldsymbol{A}^{(l)}$, given these embedding values. In general, that step is defined as

$$\boldsymbol{A}^{(l+1)} = g_l\left(\boldsymbol{H}^{(l)}, \boldsymbol{A}^{(l)}\right). \tag{5}$$

We explore two definitions for $g_l$: a general function that depends on all the nodes of the graph, $f_l$, and a convolution-based operation, $c_l$, that depends on the local connections of the graph. The former, is defined as

$$g_l\left(\boldsymbol{H}^{(l)}, \boldsymbol{A}^{(l)}\right) = f_l\left(\boldsymbol{H}^{(l)}\right) = \sigma_l\left(\tilde{f}_l\left(\boldsymbol{H}^{(l)}\right)\right), \tag{6}$$

where $\tilde{f}_l$ is approximated with a set of fully connected layers, and a non-linear function $\sigma_l$. In our experiments, we settled for two consecutive layers that reduce the encoding space before predicting the adjacency matrix for the next step, i.e., with a transformation from $\mathbb{R}^{n \times d_l} \to \mathbb{R}^{1 \times 1024} \to \mathbb{R}^{n \times n}$. (Note that this definition does not depend on the previous predicted structure, $\boldsymbol{A}^{(l-1)}$, as it relies only on the encoded information from the features of the nodes, $\boldsymbol{H}^{(l)}$. Nevertheless, for compatibility, we use the same signature for both forms of the $g_l$ function.)

The second form, $c_l$, is based on a convolution operation that processes the information of the graph locally and transforms it into a predicted adjacency. The transformation function first computes a scored adjacency, which may be interpreted as the probability of linking the nodes, defined by

$$\alpha_l\left(\boldsymbol{H}^{(l)}, \boldsymbol{A}^{(l)}\right) = \sigma_l\left(\tau\left(\boldsymbol{A}^{(l)}\right)\boldsymbol{H}^{(l)}\boldsymbol{U}^{(l)}\right), \tag{7}$$

where $\sigma_l$ is a non-linear function, $\tau(\cdot)$ is the symmetric normalization transformation (3), and $\boldsymbol{U}^{(l)} \in \mathbb{R}^{d_l \times n}$ is the learnable weight matrix for the linear combinations of the nodes' features $\boldsymbol{H}^{(l)}$. In other words, the $\alpha_l$ function broadcasts the information of the nodes' neighborhoods (as determined by the adjacency on the previous step, $\boldsymbol{A}^{(l)}$), and, at each edge, creates a score of the possible adjacency as a linear combination of the nodes' features restricted to the existing structure. Then, once again, a linear combination of the combined neighborhood's information (7) is created by

$$g_l\left(\boldsymbol{H}^{(l)}, \boldsymbol{A}^{(l)}\right) = c_l\left(\boldsymbol{H}^{(l)}, \boldsymbol{A}^{(l)}\right) = \sigma_l\left[\tau\left(\boldsymbol{A}^{(l)}\right)\boldsymbol{V}^{(l)}\alpha_l\left(\boldsymbol{H}^{(l)}, \boldsymbol{A}^{(l)}\right)\right], \tag{8}$$

where $\sigma_l$ is a non-linear function, $\alpha_l(\cdot, \cdot)$ is the approximated adjacency representation (7), and the $\boldsymbol{V}^{(l)} \in \mathbb{R}^{n \times n}$ matrix is the learnable weights that create the prediction of each edge by combining the previous scores. This operation restricts the combination of features to the local structure through the multiplication of the normalized adjacency, $\tau\left(\boldsymbol{A}^{(l)}\right)$. For the last output matrix, $\boldsymbol{A}^{(L)}$, the output

of the $c_L$ function passes through a fully connected layer of size $n \times n$, and then through a sigmoid function.

After extracting the prediction of the last layer, $\boldsymbol{A}^{(L)}$, to convert the predicted value into an edge, we use a simple threshold operation

$$A_i^o = \begin{cases} 1 & \text{if } A_i^{(L)} \geq \epsilon, \\ 0 & \text{otherwise}, \end{cases} \tag{9}$$

where $A_i^{(L)}$ is the $i$th edge value of the final prediction, $\epsilon$ is a threshold that defines what will be considered edge, and $i$ is every index of the edges in the adjacency matrix.

## 2.3 LEARNING FRAMEWORK

We are assuming that we have a family of undirected graphs, $\mathbb{G} = \{\mathcal{G}_i\}$, that have a particular structure pattern that we are interested in. We will use each of the graphs, $\mathcal{G}_i = (\boldsymbol{V}_i, \boldsymbol{A}_i)$, to learn the parameters, $\Theta$, of our model that minimize the loss function (16) on each of them. The structure of each graph is used as ground truth, $\boldsymbol{A}_i^* = \boldsymbol{A}_i$. The graph is predicted by the set of node embedding, $h_l$ (1), and the adjacency prediction, $g_l$ (5), functions that depend on a set of parameters $\Theta$, i.e., our model is defined by

$$\text{GLN} = \{h_l, g_l; \Theta\}_{l=0}^L . \tag{10}$$

Our input comprises the vertices, $\boldsymbol{H}^{(0)} = \boldsymbol{V}_i$, and some structure for training. In our experiments, we used the identity, $\boldsymbol{A}^{(0)} = \boldsymbol{I}$. However, other structures can be used as well. In the following, we describe our learning framework to obtain the parameters of the functions $h_l$ and $g_l$, for every $l$. To simplify the notation we will omit the parameters on the losses and in their functions.

Given the combinations of pairs of vertices on a graph, the total number of pairs with an edge (positive class) is, commonly, fewer than pairs without an edge (negative class). In order to handle the imbalance between the two binary classes (edge, no edge), we used the HED-loss function (Xie & Tu, 2015) that is a class-balanced cross-entropy loss function. Then we consider the edge-class objective function as

$$\mathcal{L}_{class} = -\beta \sum_{i \in Y_+} \log P\left(A_i^o\right) - (1 - \beta) \sum_{j \in Y_-} \log P\left(A_j^o\right), \tag{11}$$

where $A_i^o$ is the indexed predicted edge for the $i$th pair of vertices, where the index comes from an enumeration of the positive (edge) and negative (no edge) class of the pairs of vertices on the ground-truth graph denoted by $Y_+$ and $Y_-$, respectively; $\beta = |Y_+|/|Y|$ and $1 - \beta = |Y_-|/|Y|$ are the proportion of positive and negative pairs of vertices on the $\boldsymbol{A}^*$ graph, and $Y = Y_+ \cup Y_-$; and $P(\cdot)$ is the probability of a pair of vertices to be of a given class given by the last layer, such as

$$P\left(A_i^o\right) = A_i^{(L)}. \tag{12}$$

Individually penalizing the (class) prediction of each edge is not enough to model the structure of the graph. Hence, we compare the whole structure of the predicted graph, $\boldsymbol{A}^o$, with its ground truth, $\boldsymbol{A}^*$. By treating the edges on the adjacency matrices as regions on an image, we try to maximize the intersection over union (Rahman & Wang, 2016) of the structural regions. Then we consider the objective function,

$$\mathcal{L}_{struct} = 1 - \frac{\boldsymbol{A}^o \cap \boldsymbol{A}^*}{\boldsymbol{A}^o \cup \boldsymbol{A}^*} = 1 - \frac{\sum_{i,j} A_{i,j}^o A_{i,j}^*}{\sum_{i,j} A_{i,j}^o + A_{i,j}^* - A_{i,j}^o A_{i,j}^*}. \tag{13}$$

On the other hand, the predictions in each layer $\boldsymbol{A}^{(l)}$, where $l \in \{0, \ldots, L\}$, must be symmetric in an undirected graph. (This restriction can be removed in case of working with directed graphs, without loss of generalization.) To guarantee that, we penalize the symmetry of our predictions by a mean square loss function. Hence, we used the symmetry loss function

$$\mathcal{L}_{sym} = \frac{1}{L} \sum_{l=0}^L \left\| \boldsymbol{A}^{(l)} - \boldsymbol{A}^{(l)\top} \right\|^2, \tag{14}$$

**Table 1.** Comparison of $\text{GLN}_f$ and $\text{GLN}_c$ against deep generative models, GraphRNN, Kronecker, and MMSB, on the Community ($C = 2$ and $C = 4$), and Geometric Figures datasets. The evaluation metric is MMD for degree (Deg.), cluster (Clu.), and orbits (Orb.) shown column-wise per dataset, where smaller numbers denote better performance. For the proposed methods, the 'noise' rows denote the use of noise as input to simulate a generative method, while the others were tested on the test partition.

| | C=2 | | | C=4 | | | Geom. Figs. | | |
|---|---|---|---|---|---|---|---|---|---|
| | Deg. | Clu. | Orb. | Deg. | Clu. | Orb. | Deg. | Clu. | Orb. |
| **MMSB** | 1.7610 | 1.8817 | 1.4524 | 1.7457 | 1.9876 | 1.5095 | 0.6163 | 0.2855 | 0.6066 |
| **Kronecker** | 1.0295 | 1.2837 | 1.1846 | 1.3741 | 1.3962 | 1.3283 | 0.5817 | 0.3815 | 0.5052 |
| **GraphRNN** | 0.0027 | 0.0052 | 0.0033 | 0.2843 | 0.2272 | 1.9987 | 0.0023 | 0.0001 | 0.0015 |
| **$\text{GLN}_f$** | 0.0081 | 0.0073 | 0.7451 | 0.0021 | 0.0020 | 0.8582 | 0.0008 | 0.0002 | 0.0003 |
| **$\text{GLN}_c$** | 0.0086 | 0.0078 | 0.7395 | 0.0021 | 0.0020 | 0.8538 | 0.0014 | 0.0003 | 0.0005 |
| **$\text{GLN}_f$ on noise** | 1.1628 | 1.0938 | 1.7384 | 1.2174 | 1.0204 | 1.8807 | 0.5918 | 0.4927 | 0.5096 |
| **$\text{GLN}_c$ on noise** | 1.2095 | 1.1123 | 1.6097 | 1.2505 | 1.2276 | 1.7833 | 0.5852 | 0.4683 | 0.5134 |

where $\cdot^\top$ is the transposition operator.

We also regularize all the parameters $\Theta$ in the model by

$$\mathcal{L}_{reg} = \sum_{\theta \in \Theta} \|\theta\|^2. \tag{15}$$

Finally, we aim to minimize the total loss that is the sum of all of the previous ones, defined by

$$\mathcal{L} = \lambda_1 \mathcal{L}_{class} + \lambda_2 \mathcal{L}_{struct} + \lambda_3 \mathcal{L}_{sym} + \lambda_4 \mathcal{L}_{reg}, \tag{16}$$

where $\lambda_1$, $\lambda_2$, $\lambda_3$, and $\lambda_4$ are hyper-parameters that define the contribution of each loss to the learning process.

## 3 EXPERIMENTS

We consider two versions of our GLN model (10), one when we use a function approximator that uses the whole graph, $g_l = f_l$, that we will refer to as $\text{GLN}_f$; and the other is when we use the local operations to predict the structure, $g_l = c_l$, that we will refer to as $\text{GLN}_c$.

In this work, we evaluate our model as an edge classifier, and simulate its performance as a graph generator by inputing noise as features and predicting on them. This task is more challenging than that performed by generators, and can be considered as a lower bound for our prediction capabilities. We perform experiments on a new synthetic dataset that consists of images with geometric figures for segmentation (see Appendix A for details), and on the Community dataset that comprises two sets with $C = 2$ and $C = 4$ communities with 40 and 80 vertices each, respectively, created with the caveman algorithm (Watts, 1999), where each community has 20 people. For our experiments, we used 80% of the graphs in each dataset for training, and test on the rest.

### 3.1 ARCHITECTURE

For both models, we use the following settings. Our activation functions, $\sigma_l$, are ReLU for all layers, except for the last layer of the $g_L$ functions where $\sigma_L$ is a sigmoid. We use $L = 4$ layers to extract the final adjacency and embeddings. The feature dimensions, $d_l$, for each layer are 128, 64, 64, 3, respectively. The learning rate is set 0.0001 for all the experiments, except for the geometric figure, where the learning rate is set 0.001. Additionally, the number of epochs changes depending on the experiment. To convert the prediction of the adjacency into a binary edge, we use a fixed threshold of $\epsilon = 0.5$. The hyper-parameters in our loss function (16) are $\lambda_1 = 2$, $\lambda_2 = 10$, $\lambda_3 = 2$, and $\lambda_4 = 0.05$, for both models. In our experiments, we did not needed the regularization for the $\text{GLN}_f$, hence, $\lambda_4 = 0$ for it. Finally, for training, we used the ADAM optimization algorithm on Nvidia GTX Titan X GPU with 12 GB of memory.

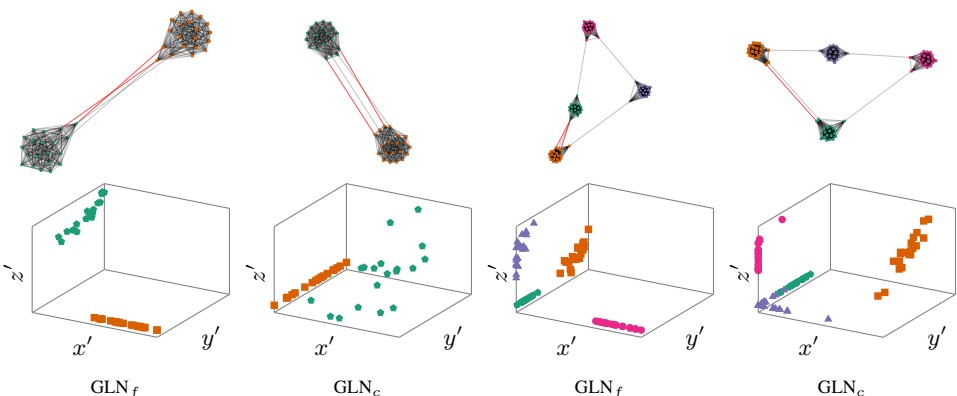

**Figure 2.** We show the predicted graph for our models and over the ground-truth of the Community dataset. On the first row, the positions of graphs' nodes correspond to the original coordinates (features), and, on the second row, we show the final 3D learned features used to predict the adjacency. The red edges represent false negatives (i.e., not predicted edges), blue edges represent false positives (i.e., additional predicted edges), and black edges are correctly predicted ones. The graphs were normalized (w.r.t. scale and translation) for better visualization.

## 3.2 GRAPH GENERATION

To evaluate the capability to learn the structure from graphs of our method, we compare it against generative graph models that also learn the structure from a given set of input graphs. We compare against traditional generative models for graphs: mixed-membership stochastic block models (MMSB) (Airoldi et al., 2008), and Kronecker graph models (Leskovec et al., 2010); and recent deep graph generative models, such as the auto-regressive model: GraphRNN (You et al., 2018). Due to infrastructure restrictions, we did not compare against generative models that have large quantity of parameters (Li et al., 2018; Simonovsky & Komodakis, 2018), and, therefore, are taxing to train. Our evaluation metric is the Maximum Mean Discrepancy (MMD) measure (You et al., 2018), which measures the Wasserstein distance over three statistics of the graphs: degree, clustering coefficients, and orbits. We report the results on this experiment on Table 1.

For the experiments on the Community dataset, for both models, we trained for 150 epochs with 400 graphs, and tested on 100. In this task, we receive as input the nodes' $(x, y)$ positions in the space. Similarly, we trained our method on the Geometric Figures dataset for 80 epochs with 2000 images, and tested on the remaining 5000. In this experiment, the inputs were the RGB information on the images.

As can be seen on Table 1, our method can predict structures over the never seen test partition for all the datasets. Additionally, we simulate a generative process by inputing noise (within the domain of the features) to the network, and analyze the structures that the method produces. We use this experiment as a way to evaluate how our method can perform on the worst case. Our results are on par with classical methods for generation of graphs (cf. Kronnecker and MMSB), but cannot improve over deep generative models on noise data.

## 3.3 EDGE PREDICTION

Our second evaluation corresponds to the accuracy of the predicted structures w.r.t. the ground truth. For this task we measure accuracy, intersection-over-union, recall, and precision. Table 2 shows our model performance on these measures.

In Fig. 2, we present our edge prediction results on the Communities using our models. Despite our model not focusing on node classification, we can clearly see a latent feature space with well separated classes. Most of the structure is recovered with few missing edges in each graph. Additional results are shown in Appendix B. Similarly, we use our models for a segmentation experiment on a synthetic dataset of Geometric Figures. Fig. 3 shows a set of graphs that divide the given images. Additional results of this experiment are shown in Appendix C.

**Table 2.** Comparison of $GLN_f$ against $GLN_c$, on the Community ($C = 2$ and $C = 4$), and with Geometric Figures datasets. The evaluation metric are accuracy (Acc.), intersection-over-union (IoU), Recall (Rec.), and Precision (Prec.) shown row-wise per method, where larger numbers denote better performance.

| | C=2 | | | | C=4 | | | | Geom. Figs. | | | |
|---|---|---|---|---|---|---|---|---|---|---|---|---|
| | Acc. | IoU | Rec. | Prec. | Acc. | IoU | Rec. | Prec. | Acc. | IoU | Rec. | Prec. |
| $GLN_f$ | 0.997 | 0.994 | 0.999 | 0.999 | 0.999 | 0.997 | 0.997 | 0.999 | 0.998 | 0.963 | 0.986 | 0.976 |
| $GLN_c$ | 0.995 | 0.991 | 0.994 | 0.999 | 0.999 | 0.994 | 0.995 | 0.999 | 0.998 | 0.974 | 0.987 | 0.980 |

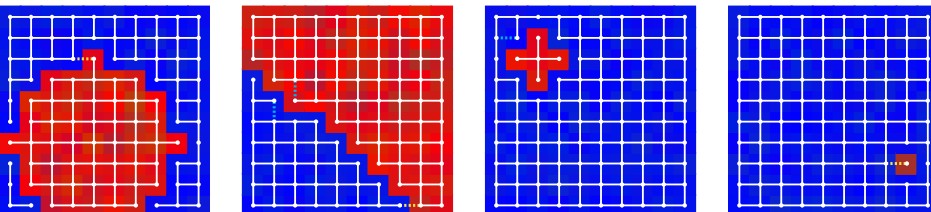

**Figure 3.** Predicted graphs using $GLN_c$ on images with geometric shape of $10 \times 10$ pixels. The image behind the graph corresponds to the input values at each node (RGB values), the white edges represent correct predictions, light blue dashed edges are false negatives (i.e., not predicted edges), and yellow dashed edges are false positives (i.e., additional predicted edges).

An interesting result of our method is that it can learn different densities of connections using the same architecture. For instance, the communities are densely connected for a given set of vertices, and then appear disconnected between the other parts. On the other hand, the geometric figures represent images that have at most four neighbors (due to the lattice structure used on the ground truth). In this case, the nodes present a constant connection rate with some of them disconnected depending on the features. Regardless of the initial input structure, the proposed methods recovered these structures without changes on their configurations.

### 3.4 ROBUSTNESS TO INITIAL STRUCTURE

We also investigated the robustness of our model to structural inputs by randomly changing the proportion of the initial connections (i.e., $10\%$, $20\%$, ..., $100\%$) in our input adjacency matrix for each input sample on the test set. (Note that the original features remained unaltered.) Fig. 4 shows the results of this experiment on the Community dataset ($C = 4$) by executing five times the generation of random structural inputs, and we are reporting the average values for both models $GLN_f$ and $GLN_c$. We obtain minimum variation on the prediction capabilities of the network. Hence, the best option is to select a minimal graph as input, i.e., the identity.

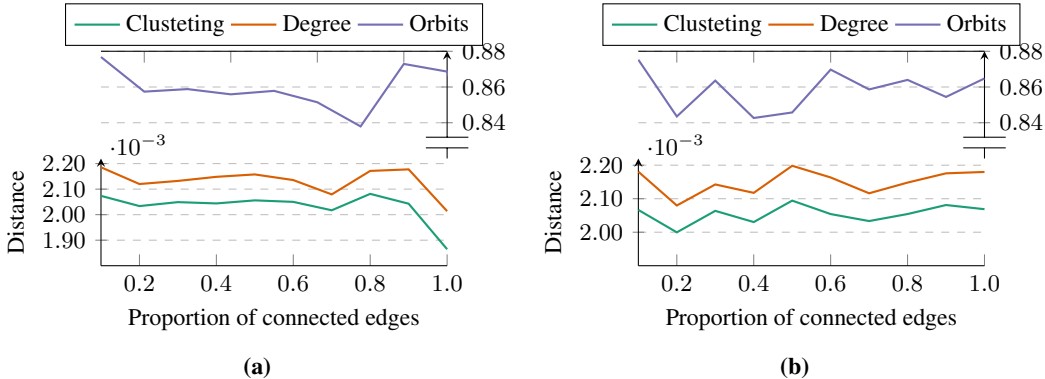

**Figure 4.** MMD metrics on $GLN_c$ (left) and $GLN_f$ (right) when varying the input structure on Community $C = 4$. The input corresponds to an adjacency matrix with different proportions of connections.

## 4 RELATED WORKS

Our approach is positioned in the center of a rich range of recent works in the areas of graph generation and classification, and link prediction. For instance, we predict the structures given a set of nodes, like link prediction, and we create rich and novel structures, like graph generation. Similarly, our node embeddings and edge prediction can be considered as a classification task.

**Link prediction's** goal is to predict the likelihood of a future relationship between two nodes in a graph. A variety of models based on Graph Convolutional Network (Kipf & Welling, 2017) have been proposed (Berg et al., 2018; Schlichtkrull et al., 2018; Zhang & Chen, 2018). For example, methods for recommendation systems on bipartite graphs were proposed by Berg et al. (2018). In addition, Schlichtkrull et al. (2018) merged auto-encoder and factorization methods (i.e., use of scoring function) to predict labeled edges. Other approaches are based on generative adversarial network (Bojchevski et al., 2018), recurrent neural networks (Monti et al., 2017), and heuristic methods (Zhang & Chen, 2018). Similarly, we predict the edges of the graph based on an initial set of nodes and a configuration. However, we learned transformations based on a neighborhood around the nodes, while also transforming the features to, in turn, enhance the structure prediction.

On the other hand, **graph classification** goal is to discriminate between different classes of graphs. The traditional methods are based on kernel graphs (Rogers & Hahn, 2010; Shervashidze et al., 2009). Usually, they calculate certain statistics on the graph structures (i.e., graph features), and then learn a classifier based on a kernel. Inspired by Convolutional Neural Networks, there is a set of methods (Defferrard et al., 2016; Duvenaud et al., 2015; Kipf & Welling, 2017; Niepert et al., 2016) that approximate convolution operations directly on the graphs. In recent years, Dai et al. (2016) and Zhang et al. (2018) tried to extract relevant features (i.e., graph embedding) from the graph structures based on the premise that groups of graphs of the same class exhibit common patterns. Currently, the models are beginning to use attention methods on the graph structures (Lee et al., 2018), allowing focus on smaller sub-structures but that contain more information. In contrast, our node embedding is driven by the structure-prediction task.

For the **generative models**, the Variational Autoencoder (VAE) (Kingma & Welling, 2014) proved to be competent at generating graphs. Generative graph VAE aims to learn a latent representation from a certain number of samples (graphs) that usually belong to the same family (Grover et al., 2018; Simonovsky & Komodakis, 2018). On the other hand, the most recent approaches combine VAE with a breadth-first search (You et al., 2018) with the objective of delimiting the search space on the graph generation. Finally, Li et al. (2018) and You et al. (2018) propose to perform auto-regressive models (i.e., generate node-to-node graphs), to generate graphs with similar structure. Despite our lack of generation from random seeds, we can simulate such process by randomly creating points and using the identity matrix to generate new graphs. Nevertheless, we consider relevant to contrast ourselves with the generative methods since they aim to learn the structures (regardless of the difference on the final task).

## 5 CONCLUSIONS

We proposed a simple yet effective method to predict the structure of a set of vertices. Our method works by learning node embedding and adjacency prediction functions, and chaining them. This process produces expected embeddings which are used to obtain the most probable adjacency. We encode this process into a neural network architecture. Our experiments demonstrate the prediction capabilities of our model on two databases with structures with different characteristics (the communities are densely connected on some parts, and sparse on others, while the images are connected with at most four neighbors). Further experiments are necessary to evaluate the robustness of the proposed method on larger graphs, with more features and more challenging structures.

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

## A  GEOMETRIC FIGURES DATASET

We made the Geometric Figures dataset for the task of image segmentation within a controlled environment. Segmentation is given by the connected components of the graph ground-truth. We provide RGB images, and their expected segmentations.

The Geometric Figures dataset contains 2500 images of size $n \times n$, that are generated procedurally.[1] Each image contains circles, rectangles, and lines (dividing the image into two parts). We also add white noise to the color intensity of the images to perturb and mixed their regions.

The geometrical figures are of different dimensions, within $[1, n]$, and positioned randomly on the image (taking care in no losing the geometric figure). There is no specific color for each geometric shape and their background.

For our experiments we use a version of dimension $n = 10$, and limit the generation for two colors.

## B  ADDITIONAL RESULTS ON COMMUNITY DATASET

We show additional results for two versions of the Community dataset on Fig. B.1.

## C  ADDITIONAL RESULTS ON GEOMETRIC FIGURES DATASETS

We show additional results for two Geometric Figures dataset on Fig. C.1.

---

[1]The code will be available after releasing the paper.

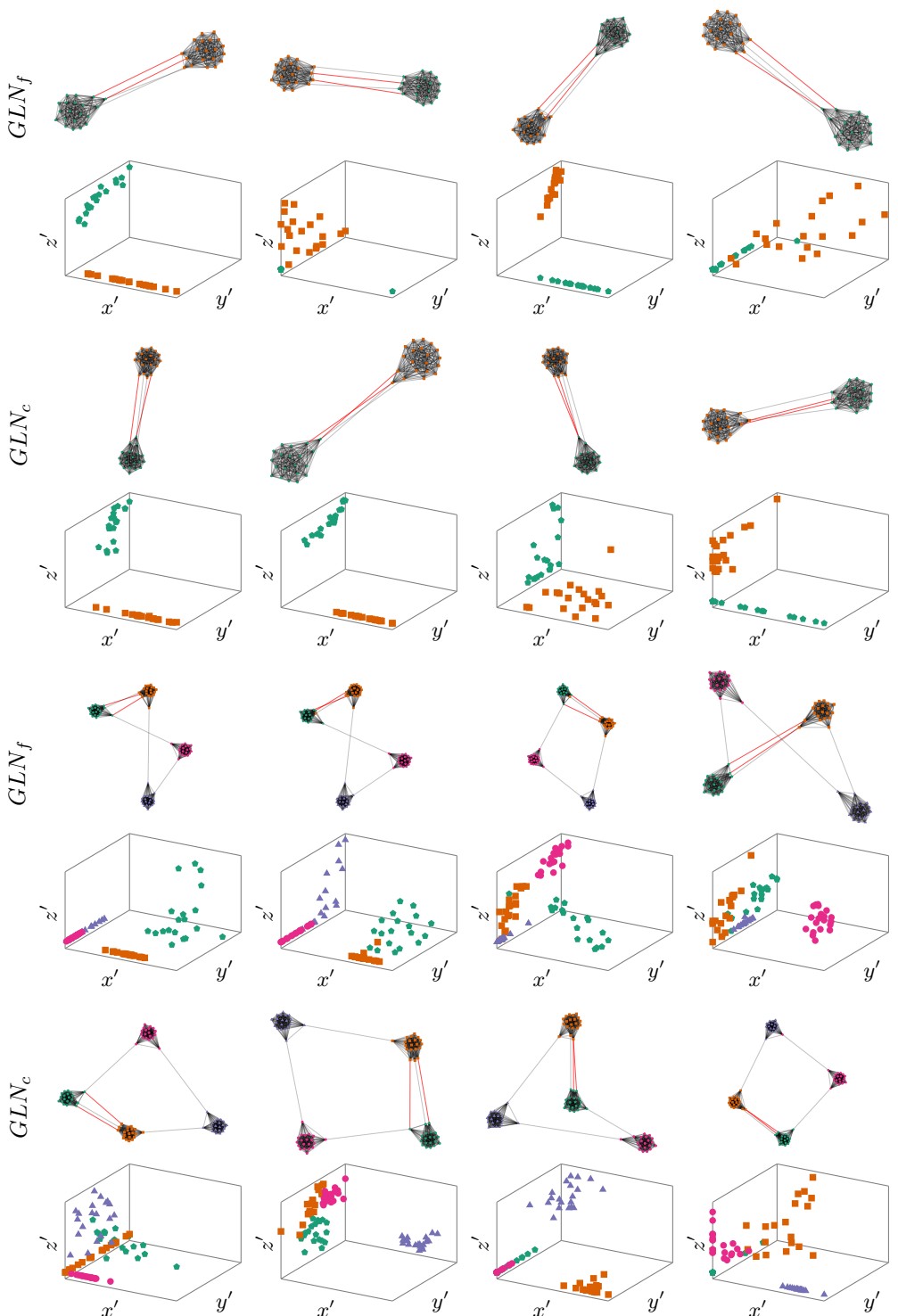

**Figure B.1.** Results on Community dataset predictions for the proposed methods, and the learned latent space, used for build adjacency matrix in the prediction. The red edges represent false negatives (i.e., not predicted edges), blue edges represent false positives (i.e., additional predicted edges), and black edges are correctly predicted ones. The graphs were normalized (w.r.t. scale and translation) for better visualization.

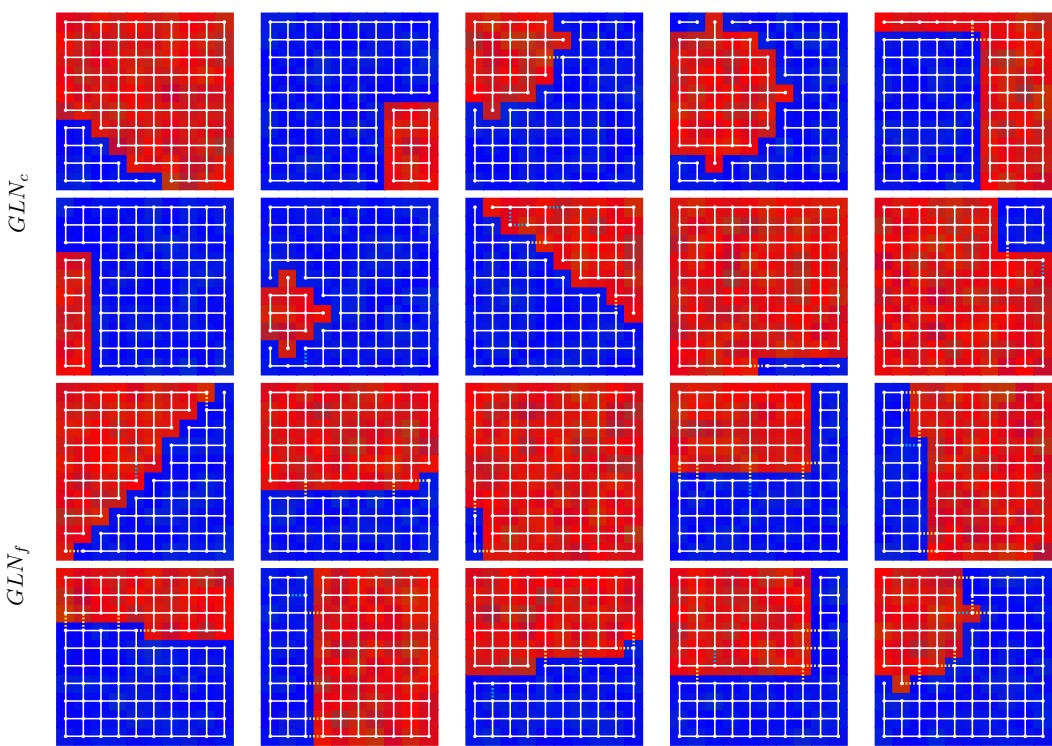

**Figure C.1.** Predicted graphs using GLN$_c$ on images with geometric shape of $10 \times 10$ pixels. The image behind the graph corresponds to the input values at each node (RGB values), the white edges represent correct predictions, light blue dashed edges are false negatives (i.e., not predicted edges), and yellow dashed edges are false positives (i.e., additional predicted edges).

