# OpenReview forum: "Graph Learning Network: A Structure Learning Algorithm"
_ICLR.cc/2019/Conference_

### Official Review · AnonReviewer2 · 2018-11-02
**Heuristic method without real data application.**

**Rating:** 4
**Confidence:** 4

**Review:**

Authors propose a supervised method for predicting adjacency matrix for a set of points. Loss function consists of 4 terms: intersection over union loss with respect to target adjacency, cross entropy loss, symmetry penalty and L2 regularization of parameters. Learning process consists of alternating node feature updates parametrized by GCN-like layers and updates of the adjacency matrix (different across layers).

My main concern is the heuristic nature of the method without any successful real data application. I do not see this work as impactful or of interest to ICLR community.

Directly regarding the content I have following comments and questions:

Word "structure" seems to be used in several meanings. For example "We consider the problem of predicting the structure of a given set of points (which we assume are the nodes of a graph) and an initial structure (connections of the points)." It only becomes somewhat clear later what is actually the learning problem studied in this paper.

"The learned convolutions" - convolution is a particular mathematical operation. I believe authors should refer to the weights of their architecture instead.

Symmetry penalty of equation 14 seems unnecessary. When optimizing for symmetric matrix it should be recognized that corresponding symmetric entries are identical variables. Hence it is sufficient, and mathematically appropriate, to correct the gradient computed without symmetric consideration. Correction is simply sum of the gradient with itself transposed (without diagonal entries).

"We compare against traditional generative models for graphs: mixed-membership stochastic block models (MMSB) " - could you please elaborate on how you use MMSB for graph generation. The use-case I am familiar with is community detection.

---

> ### Author Response · Authors · 2018-11-26
> **Reply to reviewer 2**
>
> We are grateful for the positive feedback and constructive comments from Reviewer 2.
>
> We consider that by doing the ablation, we can identify the contribution amplitude of losses in training (eq 14), but for the moment we are limited with our capacity of computational resources.

---

### Official Review · AnonReviewer1 · 2018-11-04

**Rating:** 3
**Confidence:** 4

**Review:**

This paper proposes a supervised learning method for predicting the connectivity of a graph based on both the features of nodes in the graph as well as the overall graph structure, rather than just the structure of the graph or just the node features. The approach is evaluated on two synthetic datasets, a “community” dataset and a “geometric figures” dataset.

Unfortunately, I do not think this paper as it stands currently is ready for publication at ICLR for the following reasons: (1) the comparison and discussion with prior literature is lacking; (2) the experiments are rather weak; and (3) the significance and novelty of the method itself seems limited.

1. I am very surprised there was no discussion of [1] (or even better, a comparison to), which is another method which uses information about the full graph (via message-passing) to infer the connectivity of the graph in an unsupervised way. The discussion of the literature on graph neural networks in general is a bit weak, missing important references such as [2-4]. Such approaches, especially message-passing style approaches like [4], do exactly what the current paper suggests has not been done: they make predictions based on information in the nodes while considering the structure of the graph as a whole (via message-passing). Although [4] does not explicitly make edge predictions the approach is straightforward to generalize to making edge predictions, see [5].

2. The experiments in the paper only test the proposed method on very toy domains, and thus feel weak. The results in Table 2, for example, suggest that the proposed method has reached ceiling-level performance and thus to really tell the difference between GLN_f, GLN_c, and any other methods, more difficult problems are called for. The geometric figures dataset, in particular, does not seem to me like it would test the claim that the paper would like to make: that it is important to take into account the fully structure of the graph when predicting edges. Indeed, there is a very simple rule that can be applied in the geometric figures case which does not use global graph information (if the two nodes have the same color, connect them, if they are different colors, do not connect them). It is therefore unsurprising that GLN_c actually does slightly better than GLN_f (according to Table 2) on geometric figures.

Additionally, the experiments do not provide much insight into the architecture itself. For example, the present architecture is meant to repeat the embedding and link-prediction steps some number of times, and in the experiments it seems that these steps are repeated four times. But how important is the repetition in practice? It would be nice to see the effect of repetitions on final performance, to demonstrate whether this is in fact an important component of the model or not. Similarly, there are several different loss functions but it is not obvious to what extent these losses contribute to the final performance of the model. It would be nice if there could be some ablation studies that train the model with different combinations of losses to see which are actually important.

3. Finally, I have some concerns about the model itself. If I understand correctly, both f_l and c_l depend on a number of learned parameters which is a function of the number of nodes in the graph. This is unfortunate, as one of the strengths of the graph neural network approach is that GNNs usually have a number of parameters that is independent of the size of the graph, thus allowing GNNs to scale to graphs of arbitrary size. However, that is not the case in this model. Moreover, the architecture of f_l and c_l do not seem particularly novel. f_l just involves passing the node embeddings through a MLP to produce the link predictions. c_l involves something closer to message-passing, though where weighted combinations are learned on a per-node basis (rather than sharing the same function across all nodes). This could be interesting, even though it sacrifices the scale-free nature of GNNs, if it could be shown to actually outperform existing GNN approaches on more realistic datasets. However, given the lack of experiments demonstrating this, it is hard to say how significant the approach is.

[1] Kipf, Fetaya, Wang, Welling & Zemel (2018). Neural Relational Inference for Interacting Systems. ICML 2018.
[2] Gori, M., Monfardini, G., and Scarselli, F. (2005). A new model for learning in graph domains. IJCNN 2005.
[3] Scarselli, F., Gori, M., Tsoi, A. C., Hagenbuchner, M., and Monfardini, G. (2009). The graph
neural network model. IEEE Transactions on Neural Networks, 20(1):61–80.
[4] Gilmer, J., Schoenholz, S. S., Riley, P. F., Vinyals, O., and Dahl, G. E. (2017). Neural message passing for quantum chemistry. arXiv preprint arXiv:1704.01212.
[5] Battaglia, P. W., Hamrick, J. B., Bapst, V., Sanchez-Gonzalez, A., Zambaldi, V., ... Pascanu, R. (2018). Relational inductive biases, deep learning, and graph networks. arXiv preprint arXiv:1806.01261.

---

> ### Author Response · Authors · 2018-11-26
> **Reply to reviewer 1**
>
> Dear Reviewer 1, thank you for taking the time to read and review our paper and for your useful comments.
>
> With respect to datasets, we are going behind the paper that suggested and adapt them to our task (prediction of structures) in order to make better comparisons.
> Additionally, we are analyzing message-passing approaches to increase discussions in our state-of-the-art.
>
> Concerning the amount of repetition of the steps (number of layer of the model) and the study of the different combinations of losses, we in this version, unfortunately, we do not present new results, due to the limited capacity of our resources and for the short time that we have for a new version.

---

> > ### Comment · AnonReviewer1 · 2018-11-29
> > **Response to author**
> >
> > Thanks for the response---I hope the suggested changes are useful in submitting this to another venue. As the current version stands, however, I do not feel it is ready for publication at ICLR and therefore will not be changing my score.

---

### Official Review · AnonReviewer3 · 2018-11-07
**A heuristically designed method for learning graph networks**

**Rating:** 4
**Confidence:** 5

**Review:**

This papers presents a supervised method to learn from network data. The method alternates two steps: a node embedding step (using convolutions) and an adjacency matrix update (using local convolutions or fully connected layers). These steps are stacked forming a NN that is used to represent the learning steps. The objective function is composed of a linear combination of typical losses such as cross-entropy, intersection over union and other regularization terms. The linear coefficients are treated as hyper-parameters. The methods are evaluated on graph generation and edge prediction tasks, showing results comparable to the state-of-the-art.

Overall, the paper is clearly written and addresses an important problem. However, I found the proposed method rather heuristic and not very well theoretically principled. Why should one use the proposed architecture (stacking learning steps)? What is the latent structure that this method is trying to learn, a particular sequence of graphs? Which one? Where do the supposed benefits come from? In general, both the architecture and loss (or combination of losses) need to be better justified.

Regarding experiments, on the positive side, the authors consider a representative set of methods. However, the tasks are too simple. I miss some sensitivity analysis, e.g., on the different loss functions or the number of layers. It is not clear how the method scales on the size of the networks and the depth of the layers.

minor:

- specify better how the ground truth is used in the objective
- How was the noise added? uncorrelated noise over the features?
- the loss function is referenced before being presented
- "set of node embedding" -> "set of node embedings"

---

> ### Author Response · Authors · 2018-11-26
> **Reply to reviewer 3**
>
> We thank the reviewer 3 for analyzing this paper and appreciate their pointing out important aspects.
>
> i) What is the latent structure that this method is trying to learn, a particular sequence of graphs? Which one?
> The method tries to learn intermediate representations of the final graph of output, not sequential necessarily.
>
> ii) Where do the supposed benefits come from?
> When learning intermediate representations, in each step the neighbors of the nodes change dynamically (i.e., change the neighborhood and fit the weights of each edge).
>
> iii) How was the noise added? uncorrelated noise over the features?
> The noise is non-uniform random (np.random.choice) and it is applied on the features.
>
> With respect to sensitive analysis (different number of layers and loss functions) and to use different dataset, unfortunately, we do not present results yet due to our limited capacity of resources.

---

### Meta-Review · Area_Chair1 · 2018-12-13
**A heuristic approach for graph learning, evaluated on synthetic data  only**

**Confidence:** 4
**Recommendation:** Reject

**Metareview:**

The paper addresses an important problem of supervised learning for predicting graph connectivity using both node features and the overall graph structure. The paper is clearly written, and the presented approach produces promising results on synthetic data. However, all reviewers agree that the paper could be improved by including more comparison with prior art and related work discussion, and strengthening empirical results by including real-life  data and more through evaluation; they also find the novelty and significance of the proposed approach somewhat limited. We hope the authors will use the suggestions of the reviewers to further improve the paper.